# Improving the Accuracy of Learning Example Weights for Imbalanced Classification

**Yuqi Liu & Bin Cao**[*]**& Jing Fan**
College of Computer Science, Zhejiang University of Technology, Hangzhou, China
{liuyuqi,bincao,fanjing}@zjut.edu.cn

## Abstract

To solve the imbalanced classification, methods of weighting examples have been proposed. Recent work has studied to assign adaptive weights to training examples through learning mechanisms. Specifically, similar to classification models, the weights are regarded as parameters that need to be learned. However, the algorithms in recent work use local information to approximately optimize the weights, which may lead to inaccurate learning of the weights. In this work, we first propose a novel mechanism of learning with a constraint, which can accurately train the weights and model. Then, we propose a combined method of our learning mechanism and the existing work, which can promote each other to perform better. Our method can be applied to any type of deep network model. Experiments show that compared with state-of-the-art algorithms, our method has significant improvement in varieties of settings, including text and image classification over different imbalance ratios, binary and multi-class classification.

## 1 Introduction

Classification is a fundamental task in machine learning, but in practical classification applications, the number of examples among classes may differ greatly, even by several orders of magnitude. Standard learning methods train the classification model on such an imbalanced data set, which makes the trained model biased. This bias is that the model will prefer the majority class and easily misclassify the minority class examples. This class-imbalance problem exists in many domains, such as Twitter spam detection (Li & Liu, 2018), named entity recognition (Grancharova et al., 2020) in text classification, and object detection (Oksuz et al., 2020), video surveillance (Wu & Chang, 2003) in image classification.

There are very rich research lines on using the methods of weighting examples to solve the class imbalance problem. In general, the weight of the minority class is higher than that of the majority class, so that the bias towards the majority class is alleviated. Typically, the example weight value of each class is often set to inverse class frequency (Wang et al., 2017) or inverse square root of class frequency (Mahajan et al., 2018). However, the example weights in these methods are designed empirically, hence they can not be adapted to different datasets and may perform poorly.

Recent work has studied the methods of using learning mechanisms to adaptively calculate the example weights. Ren et al. (2018) propose to use a meta-learning paradigm (Hospedales et al., 2020) to learn the weights. In this method, the example weights can be regarded as a meta-learner and the classification model is a learner. The meta-learner guides the learner to learn by weighting the example loss in the model optimization objective. More specifically, *the model objective is to get the optimal model that minimizes the example-weighted loss of the imbalanced training set.* Obviously, different weights will affect the performance of the optimal model. Which weight values make the corresponding optimal model the best? This method collects a small balanced validation set and evaluates the weight values through the validation performance of the model. Therefore, the meta-learner objective, *namely meta-objective, gives the best weights that make the optimal model minimize the loss of the balanced validation set.* This optimization problem is challenging. The key is that, in the *meta-objective*, the weights indirectly affect the loss through the optimal model, so it

---

[*]Corresponding author

is necessary to clearly define the dependence of the weights and the optimal model in the *model objective* for optimizing the weights. However, it is expensive to get this dependence through multiple gradient descent steps in the *model objective*. Ren et al. (2018) propose an online approximation method to estimate this dependence, that is, the method trains the model using a gradient descent step in the *model objective* and then can determine the relationship between the weights and the trained model in this step. Hu et al. (2019) propose to update the example weights iteratively to replace the re-estimation proposed by Ren et al. (2018), but also adopt the local approximation to optimize the weights. However, this approximation only considers the influence of the weights on the trained model in a short term (in a descent step), resulting in inaccurate learning of the weights.

In this paper, we firstly propose a novel learning mechanism that can obtain the precise relationship between the weights and the trained model in the *model objective*, so that the weights and model can be optimized more accurately. In this mechanism, we convert the *model objective* into an equation of the current model and weights. Then, we derive their relationship from this equation, and then we use this relationship to optimize the weights in the *meta-objective* and update the corresponding model. Since this optimization process always satisfies this equation, we call it learning with a constraint. However, the mechanism only uses the *model objective* to calculate the relationship but does not optimize the model for the *model objective*. To solve this problem, we integrate the method proposed by Hu et al. (2019) into our learning mechanism and propose a combined algorithm. In this algorithm, the method of Hu et al. can help to further optimize the model in the *model objective*, and our learning mechanism can make the weights and model learn more accurately. Finally, we conduct a lot of experiments to validate the effectiveness of this algorithm. The experimental settings include (1) different domains, namely text and image classification; (2) different scenarios, namely binary and multi-class classification, (3) different imbalance ratios. The results show that our algorithm not only outperforms the state-of-the-art (SOTA) method in data weighting but also performs best among other comparison methods in varieties of settings.

The remainder of this paper is organized as follows. Section 2 introduces preliminaries of the two objectives and the main idea of Hu et al. (2019). Section 3 presents our mechanism of learning with a constraint and the combined algorithm. Section 4 presents the experimental settings and evaluation results. Section 5 summarizes the related work and Section 6 concludes this paper.

## 2 PRELIMINARIES AND NOTATIONS

Let $(x, y)$ be the input and target pair. For example, in image classification, $x$ is the image and $y$ is the image label. Let $\mathcal{D}_{train}$ denote the train set, and $\mathcal{D}_{train} = \{(x_i, y_i), 1 \leqslant i \leqslant N\}$. Let $\mathcal{D}_{val}$ be a small balanced validation set, and $\mathcal{D}_{val} = \{(x_i, y_i), 1 \leqslant i \leqslant M\}$ where $M \ll N$. We denote neural network model as $\Phi(x, \theta)$, where $\theta \in R^K$ is the model parameter. The predicted value $\hat{y} = \Phi(x, \theta)$. We use loss function $f(\hat{y}, y)$ to measure the difference between predicted value $\hat{y}$ and target value $y$, and the loss function of data $x_i$ is defined as $f_i(\theta)$ for clarity. Standard training method is to minimize the expected loss on the training set: $\sum_{i=1}^{N} f_i(\theta)$, and each example has same weight. However, for an imbalanced data set, the model obtained by this method will be biased towards the majority class. Here, we aim to learn a model parameter $\theta$ that is fair to the minority class and the majority class by minimizing the weighted loss of training examples:

$$\theta^*(w) = arg\ min_\theta \sum_{i=1}^{N} w_i f_i(\theta) \tag{1}$$

where $w = (w_1, ..., w_N)^T$ is the weights of all training examples. We use $\mathcal{L}_{train}$ to represent the weighted loss on the training set $\mathcal{D}_{train}$. For a given $w$, we can obtain the corresponding optimal $\theta^*$ from Eq.1. Thus, there is a dependence between $\theta^*$ and $w$ and we write it as $\theta^* = \theta^*(w)$.

**Learning to Weight Examples** The recent work (Ren et al., 2018) proposed a method of learning the weights of training examples. In this method, the optimal $w$ is to make the model parameter $\theta^*$ obtained from Eq.1 minimize the loss on a balanced validation set. It means that this model performs well on a balanced validation set, and it can fairly distinguish examples from different classes. Formally, the optimal $w$ is given as

$$w^* = arg\ min_w \frac{1}{M} \sum_{i=1}^{M} f_i^v(\theta^*(w)) \tag{2}$$

where the superscript $v$ stands for validation set. Let $\mathcal{L}_{val}$ be the loss on the validation set $\mathcal{D}_{val}$.

**Learning the Parameters** The recent work (Hu et al., 2019) introduced an algorithm of solving the model parameter $\theta^*$ and weight $w^*$. The algorithm optimizes $\theta$ and $w$ alternately until convergence. In each iteration, the algorithm utilizes a gradient descent step in Eq.1 to approximate the relationship between $\theta$ and $w$, and then calculates the gradient $\nabla_w \mathcal{L}_{val}$ and $\nabla_\theta \mathcal{L}_{train}$ to update $w$ and $\theta$ respectively.

More specifically, at the $t$-th iteration, the algorithm first calculates the approximate relationship between $\theta$ and $w$ through the $t$-th gradient descent step in Eq.1. We define a matrix $F(\theta) = (\nabla f_1(\theta), ..., \nabla f_N(\theta))$, whose column vector represents the derivative of $f_i(\theta)$ with respect to $\theta$, so we calculate the derivative of $\nabla_\theta \mathcal{L}_{train}$ with respect to $\theta$ as $\nabla_\theta \mathcal{L}_{train} = F(\theta)w$. Then, the $t$-th gradient descent step of $\theta$ is given as

$$\hat{\theta}_{t+1} = \theta_t - \eta_\theta F(\theta_t)w_t \tag{3}$$

where $\eta_\theta$ is the descent step size on $\theta$. In order to avoid very expensive calculations, the algorithm ignores the influence of $w$ on $\theta_t$. Therefore, in the single gradient descent step, $\hat{\theta}_{t+1}$ linearly depends on $w$.

Then, based on this linear dependence, the algorithm can calculate the gradient $\nabla_w \mathcal{L}_{val}$ and uses gradient descent to update $w$, and then updates $\theta$ again to make it perform better on validation set. Substituting the updated $\hat{\theta}_{t+1}$ into Eq.2, we have $\mathcal{L}_{val} = \frac{1}{M}\sum_{i=1}^{M} f_i^v(\hat{\theta}_{t+1}(w))$. We can observe that the $w$ acts on $\hat{\theta}_{t+1}$ and then affects $\mathcal{L}_{val}$. Thus, combining Eq.3, we can calculate the gradient $\nabla_w \mathcal{L}_{val} = (\nabla_w \hat{\theta}_{t+1})^T \nabla_{\hat{\theta}_{t+1}} \mathcal{L}_{val} = -\eta_\theta F(\theta_t)^T \nabla_{\hat{\theta}_{t+1}} \mathcal{L}_{val}$, so the update on $w$ at $t$ step is given as

$$w_{t+1} = w_t + \eta_w \eta_\theta F(\theta_t)^T \nabla_{\hat{\theta}_{t+1}} \mathcal{L}_{val} \tag{4}$$

where $\eta_w$ is the descent step size on $w$. According to gradient descent theory, when $\eta_w$ is appropriately small, $\mathcal{L}_{val}(w_{t+1}) \leq \mathcal{L}_{val}(w_t)$. This means using $w_{t+1}$ to update $\theta$ performs better than $w_t$. Therefore, the algorithm substitutes the updated $w_{t+1}$ into Eq.3 and obtain the new update on $\theta$

$$\theta_{t+1} = \theta_t - \eta_\theta F(\theta_t)w_{t+1} \tag{5}$$

where $\theta_{t+1}$ satisfies $\mathcal{L}_{val}(\theta_{t+1}) \leq \mathcal{L}_{val}(\hat{\theta}_{t+1})$, that is, $\theta_{t+1}$ have better validation performance than $\hat{\theta}_{t+1}$.

Finally, the algorithm repeatedly calculates Eq.3, 4 and 5 and alternately optimizes $\theta$ and $w$ until convergence.

## 3  New Method of Learning the Parameters

In this section, we introduce a new method to learn the model parameter $\theta^*$ and weight $w^*$ in Eq.1 and Eq.2. First, in Section 3.1, we propose to learn $\theta$ and $w$ with a constraint, which can accurately optimize $\theta$ and $w$. Then, in Section 3.2, we propose a combined method to train $\theta$ and $w$ to make the model parameter $\theta$ have better performance.

### 3.1  Learning With a Constraint

In the section, we first analyze the difficulty of solving $\theta^*$ and $w^*$. Gradient-based optimization is a commonly used method in machine learning. Thus, we first need to calculate the gradient $\nabla_\theta \mathcal{L}_{train}$ and $\nabla_w \mathcal{L}_{val}$. Based on Eq.2, we have $\nabla_w \mathcal{L}_{val} = (\nabla_w \theta^*)^T \nabla_{\theta^*} \mathcal{L}_{val}$. However, it is difficult to explicitly give the form of function $\theta^*(w)$, resulting in $\nabla_w \mathcal{L}_{val}$ cannot be calculated directly. The previous work obtained the relationship between $\theta$ and $w$ through the gradient descent process of $\theta$, and only considered the influence of $w$ on $\theta$ in a single gradient descent step. Based on this relationship, calculating the gradient and updating the parameter is not precise.

Here, we obtain the relationship between $\theta$ and $w$ from a new perspective. First, we observe the gradient $\nabla_\theta \mathcal{L}_{train}$, that is,

$$\nabla_\theta \mathcal{L}_{train} = F(\theta)w = c \tag{6}$$

---

**Algorithm 1:** Learning to Weight Examples Using a Combination Method

---

**Input** : The network model parameter $\theta$
The weight of training examples $w$
Training set $\mathcal{D}_{train}$; Validation set $\mathcal{D}_{val}$
The number of iterations of the combination method $T$
The number of iterations of our method $T'$

1 Initialize model parameter $\theta$ and weight $w$
2 **for** $t = 0 \ldots T - 1$ **do**
3     Calculate the relationship between $\theta$ and $w$ on $\mathcal{D}_{train}$ through Eq.3
4     Optimize $w$ on $\mathcal{D}_{val}$ through Eq.4
5     Update $\theta$ through Eq.5
6     **for** $t' = 0 \ldots T' - 1$ **do**
7         Calculate the derivative $\nabla_w \theta$ on $\mathcal{D}_{train}$ through Eq.7
8         Optimize $w$ on $\mathcal{D}_{val}$ through Eq.8
9         Update $\theta$ through Eq.9

**Output:** Trained model parameter $\theta^*$ and weight $w^*$

---

where $c$ is the gradient value. We can see that changing the value of $w$ can find corresponding $\theta$ to satisfy Eq.6. It means that there is a functional relationship between $\theta$ and $w$ in Eq.6. Because all $\theta$ and $w$ satisfying this equation have the same value of $\nabla_\theta \mathcal{L}_{train}$, we also call Eq.6 a constraint of $\theta$ and $w$. In particular, the optimal model parameter $\theta^*$ and $w$ satisfy the constraint: $F(\theta^*)w = 0$. Then, we can make use of the constraint to derive a precise relationship between $\theta$ and $w$. Our network model may be very complex, and we cannot explicitly give the functional form of $\theta$ and $w$ according to the constraint. However, by applying the implicit function theorem, the derivative of $\theta$ with respect to $w$ in Eq.6 can be obtained as follow

$$\nabla_w \theta = -[\nabla_\theta(F(\theta)w)]^{-1} F(\theta) = -H^{-1}F(\theta)$$

where $H \in R^K \times R^K$ is the Hessian matrix, namely, the second derivative of $\mathcal{L}_{train}$ with respect to $\theta$. However, calculating an exact Hessian matrix is very expensive. Especially nowadays network models have a huge amount of parameters. In addition, in this case, we require the inverse of $H$, rather than $H$ itself. Therefore, we adopt diagonal approximation to evaluate $H$ (Bishop, 2006). In other words, we only need to calculate the diagonal elements of $H$. Furthermore, it is trivial to calculate the inverse by taking the reciprocal of the diagonal elements. Let $h \in R^K$ be the reciprocal of the diagonal elements of $H$. Then, the derivative is evaluated as

$$\nabla_w \theta = -diag(h)F(\theta) \tag{7}$$

Next, we can make use of the derivative to calculate the gradient $\nabla_w \mathcal{L}_{val}$, and then update $w$ and $\theta$. The update process always satisfies the constraint of Eq.6, so we call it learning with a constraint. Combining Eq.7, we have $\nabla_w \mathcal{L}_{val} = -F(\theta)^T diag(h)\nabla_\theta \mathcal{L}_{val}$. Thus, the update of $w$ is

$$w' = w + \eta'_w F(\theta)^T diag(h)\nabla_\theta \mathcal{L}_{val} \tag{8}$$

where $\eta'_w$ is the step size. Then, we use the updated $w'$ to calculate the corresponding $\theta'$ in the constraint. Since we do not know the explicit functional form of $\theta$ and $w$ in Eq.6, we use the first order derivative to approximate $\theta'$. Combing Eq.7, $\theta'$ is evaluated as

$$\theta' \approx \theta + \nabla_w \theta(w' - w) = \theta - diag(h)F(\theta)(w' - w) \tag{9}$$

Finally, under the condition of satisfying the constraint, we repeatedly optimize $w$ and $\theta$, corresponding to Eq.8 and Eq.9, until convergence. The detailed proof of this convergence can be found in Theorem 1 in Appendix A.2.

## 3.2 LEARNING IN A COMBINED WAY AND IMPLEMENTATION

The method we proposed in Section3.1 still has a shortcoming. In the method, we make use of the gradient $\nabla_\theta \mathcal{L}_{train}$ of Eq.1 to obtain a constraint $\nabla_\theta \mathcal{L}_{train} = c$, and then calculate the solution of $w$ and $\theta$ under the constraint. However, this method only ensures that the solution is optimal in Eq.2

under the constraint, and cannot guarantee that the solution of $\theta$ is optimal in Eq.1. Because our method only use Eq.1 to obtain the constraint, but not to optimize $\theta$ for Eq.1, and when $c \neq 0$, the solution of $\theta$ is not optimal in Eq.1.

In order to calculate the better solution, our method needs to be combined with another algorithm that can make $\theta$ reach the optimal in Eq.1. The method of Hu et al. in Section2 is a more appropriate choice, rather than directly updating $\theta$ using the gradient $\nabla_\theta \mathcal{L}_{train}$. Because it will first adjust $w$ and then update $\theta$ based on the new $w$. It is explained in Section2 that $\theta$ obtained in this way has a better validation performance than $\theta$ directly using gradient descent.

Therefore, we propose a way to learn $\theta$ and $w$ by combining our method in Section3.1 with the method (Hu et al.) in Section2. In this way, we alternately use these two methods to learn $\theta$ and $w$. In each iteration, we first update $\theta$ and $w$ using the method (Hu et al.). It can make $\theta$ reduce the value of $\mathcal{L}_{train}$ and approach the optimal in Eq.1, while $\theta$ also reduces $\mathcal{L}_{val}$ and perform well on validation set. Then, we optimize $\theta$ and $w$ using our method until convergence, so that $\theta$ further reduces $\mathcal{L}_{val}$ and has the best verification performance among all $\theta$ with the same gradient $\nabla_\theta \mathcal{L}_{train}$.

This combined way can overcome the shortcomings of each method. On the one hand,the method (Hu et al.) can use the gradient $\nabla_\theta \mathcal{L}_{train}$ to update $\theta$ and make $\theta$ close to the optimal in Eq.1. It makes up for the shortcoming that our method cannot optimize Eq.1. On the other hand, the method (Hu et al.) only considers the influence of $w$ on $\theta$ in a single gradient descent step, and then uses this approximation to optimize Eq.2. Hence, $\theta$ obtained by the method (Hu et al.) may not be optimal for Eq.2. Our method can make use of a constraint to derive an accurate functional relationship between $\theta$ and $w$. Thus, by optimizing $\theta$ through our method, a better solution can be obtained.

This combination algorithm is listed in Algorithm 1. It takes $T$ iterations to alternately use two methods to optimize $\theta$ and $w$. In $t$-th iteration, it first adopts the method (Hu et al.) to update $w$ and $\theta$ (lines 3-5), and then it uses our method to optimize $w$ and $\theta$ repeatedly T times (lines 6-9), making $\theta$ converges under the current constraint. Finally, it outputs the trained model parameter $\theta^*$ and weight $w^*$. The proof of convergence of the Algorithm 1 can be found in Theorem 2 in Appendix A.2. In addition, we discuss the convergence rate of Algorithm 1. According to the conclusion in the paper (Ren et al., 2018), when we take $T$ steps to update the parameter $\theta$ through the method (Hu et al.), it can achieve $\| \nabla_\theta \mathcal{L}_{val} \| \leq \mathcal{O}(\frac{1}{\sqrt{T}})$ where $\| \nabla_\theta \mathcal{L}_{val} \|$ is the update precision of parameter $\theta$. For the method in Section 3.1, achieving the same precision requires $\mathcal{O}(\sqrt{T})$ steps. More detailed proofs are in Theorem 3 in Appendix A.3. Therefore, as the combined method, Algorithm 1 needs $T \times \mathcal{O}(\sqrt{T}) = \mathcal{O}(T^{\frac{3}{2}})$ to converge.

## 4 EXPERIMENTS

In this section, we perform extensive experiments to validate the effectiveness of our method. First, we describe the experimental setup in detail. Second, we compare different methods in two domains: text and image classification and in two situations: binary classification and multi-class classification. Third, we design experiments to study the performance of our method in different imbalance ratios. Moreover, we evaluate the performance of our methods with different metrics on a large-scale data set in Appendix A.1.

### 4.1 EXPERIMENTAL SETUP

**Models.** We choose two network models for text and image classification. Specifically, in text classification, we use the BERT (base, uncased) model (Devlin et al., 2018) to extract the 768-dimensional representation of text data (Xiao, 2018) and then use a simple 4-layer fully connected network (FCN) for classification. The FCN model is given as Table 1. The pair of numbers in brackets respectively indicate the sizes of input and output of the linear layer. In addition, the first two layers apply rectified linear unit activation (ReLU) function to avoid the vanishing gradient problem during training, and the third layer uses a nonlinear activation function (Tanh) to enhance the model learning ability. The last layer is the classification layer, in which the size of output depends on the number of classes in a classification task. In image classification, we use the ALL-CNN-C network model that is a sequence of 9 convolution layers. Noting that our method does not rely on the classification model, and can also be applied to other models.

Table 1: The network model for text classification

| Input 768-dimensional text representation |
| :---: |
| (768, 768) linear layer, ReLU |
| (768, 768) linear layer, ReLU |
| (768, 10) linear layer, Tanh |
| (10, size of labels) linear layer |

Table 2: Description of four data sets

| Data sets | Classes | Fine-tune / Pretrain |
| :--- | :---: | :---: |
| SST-2 | 2 | $2 \times 500$ |
| SST-5 | 5 | $5 \times 500$ |
| CIFAR10 | 10 | $10 \times 4000$ |
| CIFAR100 | 2 | 0 |

**Data Sets and Model Preparation.** We choose 4 data sets for text and image classification, and we use part of training examples in the data sets to prepare for the subsequent training of the models. The information of the four data sets we used is shown in Table 2. In text classification, we use two popular benchmark datasets. We use the SST-2 sentiment analysis benchmark (Socher et al., 2013) for binary classification, and use the SST-5 sentence sentiment (Socher et al., 2013) with 5 categories for multi-class classification. In image classification, we adopt the commonly-used CIFAR10 (Schneider et al., 2019) for multi-class classification experiment and select the examples of class 0 and 1 from CIFAR100 (Schneider et al., 2019) to form a data set for binary classification.

To make subsequent experiments on strong models, we use part of training examples to fine-tune the BERT and pre-train the ALL-CNN-C model respectively. In text classification, we use the text data in a specific domain to fine-tune BERT, so that we can extract the better text representations from the fine-tuned BERT and improve the performance of the FCN model. In image classification, we first pre-train the ALL-CNN-C model using image data, and then the pre-trained model can be helpful to improve downstream tasks for subsequent experiments. In Table 2, we list the number of these examples. On SST-2 and SST-5 data sets, we take out 500 training examples of each class to fine-tune the BERT model. For the experiments on CIFAR10, we use 4000 training examples per class to pre-train the ALL-CNN-C model. In the experiments on CIFAR100, we do not pre-train the model, because it can perform well on this binary classification task without pre-training. Noting that the training examples used to improve the models will not be used in subsequent experiments.

**Comparison Methods.** We compare our method with five approaches: (1) Baseline, a method without any processing. In other words, the classification model is directly trained on an imbalanced training set. (2) Proportion, a commonly used method that weights examples by inverse class frequency. (3) Hu et al.'s, is the SOTA approach (Hu et al., 2019) for data weighting, which is described in Section 2 and implemented using the code[1] provided by the authors. In addition, since we set a small validation set in our experiments, the methods that need to be learned on the validation set are easy to over-fit. Therefore, in the following methods, we add regularization for the model parameters in Eq.2. (4) Hu et al.'s+R, a method that adds regularization to the validation learning in the method (Hu et al.). (5) Two-phase (Wahab et al., 2017), a learning method divided into two phases. It first trains the model to learn a good classification representation on an imbalanced training set and then adjusts the imbalance bias of the model by learning on a balanced validation set. When the model is trained on the validation set, we also add regularization.

**Training and Evaluation.** In our experiments, we first fine-tune the BERT model and pre-train the ALL-CNN-C model. In the following training, the text data is first converted into vector representations by BERT and then used to train the FCN model, and when training on CIFAR10, the ALL-CNN-C model is initialized by the pre-trained model. Next, we apply different methods to train the models. We divide this training process into 2 stages, and we take an imbalanced training set and a small balanced validation set from the remaining training examples (not including the examples used for model preparation). In Stage 1, we only use the training set to train the models, and the trained models can be regarded as the model initialization for subsequent training. For the method (Hu et al.), the trained model has basic classification capabilities, so that it can use stable gradient information to optimize the weights during weighting the examples (Ren et al., 2018). For the two-phase method, Stage 1 corresponds to its first learning phase. In Stage 2, we train the models according to their respective methods. Our method and the method (Hu et al.) will learn the model and example weight using the training set and validation set together. For the two-phase method, we only train the model on the balanced validation set, corresponding to its second phase. For Baseline and Proportion, the models still learn on the training set.

---

[1]Code available at https://github.com/tanyuqian/learning-data-manipulation

Table 3: Settings of the training process on 4 data sets

| Data sets | Fine-tune / Pretrain | Stage 1 | Stage 2 |
|---|---|---|---|
| SST-2 or SST-5 | Adam(5e-6) epochs:5 batch size:8 | Adam(1e-2) epochs:15 batch size:50 | Adam(1e-2) epochs:10 batch size:50 |
| CIFAR10 | follow the training (Springenberg et al., 2014) | Adam(1e-6) epochs:200 batch size:128 | Adam(1e-5) epochs:10 batch size:128 |
| CIFAR100 | - | follow the training (Springenberg et al., 2014) | Adam(1e-4) epochs:10 batch size:128 |

Table 4: Results of six methods on four data sets

| Methods | SST-2 100:1000 | SST-5 50:500 | CIFAR10 50:500 | CIFAR100 40:400 |
|---|---|---|---|---|
| Baseline | $75.52 \pm 2.99$ | $40.24 \pm 0.99$ | $69.95 \pm 3.35$ | $85.00 \pm 1.10$ |
| Proportion | $79.59 \pm 3.35$ | $42.59 \pm 1.12$ | $79.58 \pm 0.34$ | $85.20 \pm 1.21$ |
| Two-phase | $81.99 \pm 0.80$ | $42.60 \pm 1.44$ | $79.63 \pm 0.44$ | $86.00 \pm 1.61$ |
| Hu et al.'s | $81.57 \pm 0.74$ | $39.82 \pm 1.07$ | $79.36 \pm 0.51$ | $85.40 \pm 1.07$ |
| Hu et al.'s+R | $82.25 \pm 1.16$ | $40.14 \pm 0.39$ | $79.55 \pm 0.21$ | $86.50 \pm 2.41$ |
| **Ours.** | $\mathbf{82.58 \pm 0.98}$ | $\mathbf{44.62 \pm 1.08}$ | $\mathbf{79.71 \pm 0.37}$ | $\mathbf{87.40 \pm 1.66}$ |

The settings of the training process on the four data sets are listed in Table 3. Each cell in the table indicates the settings in the current stage, including the optimizer used, learning rate, number of epochs, and batch size, where the value in brackets is the learning rate. In text classification, we use Adam optimization. In image classification, we first follow the implementation of Springenberg et al. to use SGD optimization, and then we use Adam to optimize in subsequent training.

Finally, we indicate the evaluation criteria and hyperparameters tuning. We use the accuracy on the full test set of each data set to evaluate the performance of models. During the training process, the final result may be over-fitting, so we record the best step corresponding to the highest accuracy on the validation set. In addition, we also tune a series of hyperparameters for different methods and report the best in the test set. For the method (Hu et al.), we follow (Hu et al., 2019) and set the decay of weight to avoid exploding value. The decay value is selected from $\{1, 5, 10\}$. For Hu+regularization, we set the learning rate for weight update, which is taken from $\{1, 1e-1, 1e-2, 1e-3\}$. For our method, we set the learning rate and epochs for updating the weights during learning with constraint, and they are taken from $\{1e-2, 1e-3, 1e-4, 1e-5\}$ and $\{1, 5, 10, 15, 30\}$ respectively. We adopt general regularization, namely $L_p$-norm (Bohra & Unser, 2020), for the methods that need to be trained on the validation set. The value of $p$ is selected from $\{2, 4, 6, 8\}$. The log value of regularization coefficient is selected from $\{-4, ..., 4\}$ for text data sets and $\{-4, ..., 9\}$ for image data sets. All experiments were implemented with Python 3.8 and PyTorch 1.8 and were evaluated on a Linux server with RTX 3080 GPU and 128GB RAM. All results are averaged over 5 runs $\pm$ one standard deviation.

## 4.2 RESULTS ON DIFFERENT DATA SETS

We compare the performance of different methods on the four data sets. The four data sets involve text and image domains, as well as binary classification and multi-class classification scenarios. They can more comprehensively reflect the performance of our method. In this experiment, we set an imbalance ratio of 1:10, which is the ratio of the example size of the minority class to the majority class. In all data sets, we set class 0 as the minority class, and the rest as the majority class. On the four data sets, the size of training examples is different. We set the number of training examples for each majority class of the data set SST-2, SST-5, CIFAR10, CIFAR100 to 1000, 500, 500, 400 respectively. In addition, for all data sets, the number of examples in the validation set is 10 for each class. The training set and validation set are randomly selected from the remaining training examples in each data set.

Table 5: Results of different imbalance ratios on SST-2 data set

| Methods | 10:1000 | 20:1000 | 100:1000 |
|---|---|---|---|
| Baseline | 49.92 ± 0.00 | 49.92 ± 0.00 | 75.52 ± 2.99 |
| Proportion | 60.63 ± 13.13 | 78.76 ± 2.40 | 79.59 ± 3.35 |
| Two-phase | 75.35 ± 8.90 | 80.52 ± 1.96 | 81.99 ± 0.80 |
| Hu et al.'s | 55.84 ± 11.84 | 73.61 ± 11.86 | 81.57 ± 0.74 |
| Hu et al.'s+R | 66.68 ± 13.99 | 79.53 ± 1.64 | 82.25 ± 1.16 |
| **Ours.** | **80.62 ± 0.93** | **81.14 ± 1.25** | **82.58 ± 0.98** |

Table 6: Results of different imbalance ratios on CIFAR100 data set

| Methods | 4:400 | 8:400 | 40:400 |
|---|---|---|---|
| Baseline | 64.40 ± 11.98 | 77.40 ± 12.23 | 85.00 ± 1.10 |
| Proportion | 60.50 ± 8.40 | 69.60 ± 8.56 | 85.20 ± 1.21 |
| Two-phase | 66.00 ± 13.95 | 79.50 ± 3.75 | 86.00 ± 1.61 |
| Hu et al.'s | 60.20 ± 8.19 | 69.10 ± 8.08 | 85.40 ± 1.07 |
| Hu et al.'s+R | 71.80 ± 11.73 | 82.50 ± 4.27 | 86.50 ± 2.41 |
| **Ours.** | **77.20 ± 3.75** | **82.60 ± 3.87** | **87.40 ± 1.66** |

**Results** The results on the four data sets are shown in Table 4. We can see that our method has the best performance in these 4 data sets. Especially on the SST-5 data set, our method exceeds the second-best method by more than 2 accuracy points. It demonstrates that our method can perform well in multiple domains and different classification scenarios. Hu et al.'+R and Two-phase are competitive methods. On the SST-2 and CIFAR100, Hu et al.'+R is the second-best, and on the SST-5 and CIFAR10, Two-phase also reaches the second-best. It shows that using a balanced data set to simply adjust a biased model can also achieve good results. In addition, Hu et al.'+R performs better than Hu et al.'s. on all data sets, and on the CIFAR100, it surpasses the latter by more than 1 accuracy point. It indicates that adding regularization to the validation learning can effectively improve the method (Hu et al.). However, on the SST-5, the method (Hu et al.) performs worse than the baseline, which may be due to the ineffectiveness of the approximation on SST-5. The accuracy of the proportion method is lower than that of our method by more than 2 accuracy points on the SST-2, SST-5, and CIFAR100. It shows that the method of learning weight has more advantages than weighting empirically. The baseline method performs the worst on three data sets due to the lack of measures to solve the imbalance.

## 4.3 RESULTS OF DIFFERENT IMBALANCE RATIOS

We study the performance of our method with different imbalance ratios. In this experiment, we use the SST-2 and CIFAR100 data sets, and we vary the imbalance ratio from {1:10, 1:50, 1:100}. The example size of majority classes in the training set and the validation set are consistent with the setting of Section 4.2. In addition, the training set and validation set are also randomly constructed.

**Results** Table 5 and Table 6 respectively shows the results of different imbalance ratios on SST-2 and CIFAR100 data set. The results are listed in the order of imbalance ratios of 1:10, 1:50, 1:100. There are three main observations. First, our method has achieved the highest accuracy rates in all imbalance ratio settings. It further demonstrates that our method can have excellent performance in different situations, such as slight imbalance, extreme imbalance, etc. Second, as the training data becomes more imbalanced, the performance of our method is more dominant than other methods. On the SST-2, the accuracy of our method exceeds the second-best method by about 0.3 at 100:1000 and more than 5 accuracy points at 10:1000. Similarly, on the CIFAR100, the accuracy of our method improves the second-best method over 0.9 at 100:1000 and more than 6 accuracy points at 10:1000. It shows that our method is more advantageous in extreme imbalance. Third, when the imbalance ratios are 1:50 and 1:100, the accuracy rates of the proportion method are almost lower than other imbalance classification methods. On the CIFAR100, the proportion method even performs worse than the baseline. It indicates that as the data imbalance becomes serious, the proportion method may not be effective. On the contrary, the advantage of the methods of learning weights is more obvious.

## 5 RELATED WORK

There have been very rich studies on weighting examples for imbalance classification, and these studies can be grouped into two categories, namely *empirical weighting and automatic weighting.*

**Empirical Weighting.** The empirical weighting methods assign the manual weight values to the examples. Generally, the minority class example will be assigned a larger weight value than that of the majority class, so as to relieve the bias of the model trained on the imbalanced data set. The methods of weighting by class are first proposed (King & Zeng, 2001). In these methods, the examples of each class are manually set to the same value, such as inverse class frequency (Wang et al., 2017; Huang et al., 2016) or inverse square root of class frequency (Mikolov et al., 2013; Mahajan et al., 2018). Cui et al. (2019) proposed to calculate the effective number of examples as class frequency and then also use its inverse to weight examples and achieved significant improvements on long-tailed training data. In addition, many methods of weighting by example have also been proposed. Hard example mining (Shrivastava et al., 2016) thought focusing on the hard examples can improve the model on the imbalanced data. Dong et al. (2017); Malisiewicz et al. (2011) proposed to utilize the example loss to find hard examples and assign them higher weights. Lin et al. (2017) proposed to use the predicted probability to calculate higher weights for the hard examples and dynamically adjust the weight values during training. Empirical weighting is convenient to implement and can achieve excellent performance, though it cannot adapt to different data sets and may cause poor performance. In addition, manually setting weights will also increase the engineering burden.

**Automatic Weighting.** The automatic weighting methods assign adaptive weights to the examples through learning mechanisms. Curriculum learning can provide an example weighting strategy for neural network models to learn on corrupted labels (Jiang et al., 2018; Wei et al., 2021), but the method focuses on examples that are easy to learn (Zhang et al., 2020). On the contrary, imbalance learning prefers hard examples, so it is different from the methods of data weighting in imbalance classification. Ren et al. (2018) proposed to learn the example weights by a meta-learning paradigm (Zhang et al., 2021). This algorithm treats the example weights as a meta-learner and guides the learner to learn on the imbalanced training set. The loss on the balanced validation set is used as the meta-objective to optimize the example weights (Bai et al., 2021). In each iteration of updating the weights, this algorithm uses a gradient descent step to approximate the relationship between the weights and the learner. Hu et al. (2019) improved the algorithm by iteratively optimizing weights instead of re-estimation at each iteration. Our work is based on the work of Ren and Hu et al. and make further research. There is a key difference between our work and theirs. We use the model optimization objective to derive the precise relationship between the model and weights, instead of the local approximation strategy they used. Therefore, our algorithm can accurately optimize the example weights and get a better model for imbalance classification. The massive experimental results show that our algorithm makes significant improvements.

## 6 CONCLUSION

In this paper, based on the work of Ren and Hu et al., we further propose an improved algorithm to learn the example weights for imbalance classification. In this algorithm, we propose a learning mechanism that can accurately update the weights and the classification model under a constraint and improves the validation performance of the model. This is a key improvement compared to the method proposed by Ren et al. that uses the local approximation to optimize the weights. In addition, the algorithm we proposed is a combination of our learning mechanism and the method proposed by Hu et al., which can promote each other and make the model perform better. Finally, the experimental evaluation shows that our algorithm can achieve significant improvement compared with the SOTA method in data weighting and other imbalance methods. In our future work, we plan to extend our algorithm and explore the performance of our algorithm in data augmentation.

**Acknowledgements.** This research was partially sponsored by the following funds: National Key R&D Program of China (2018YFB1402800), Key Research Project of Zhejiang Province (2022C01145), Fundamental Research Funds for the Provincial Universities of Zhejiang (RF-A2020007) and Zhejiang Lab (2020AA3AB05).

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

## A APPENDIX

### A.1 EXPERIMENTAL RESULTS WITH DIFFERENT METRICS

In this section, we evaluate the performance of our methods with different metrics on a large-scale data set. In addition to the accuracy, we also adopt the Marco-F1 score and G-means that are often used in imbalanced classification. In this experiment, we use an additional large-scale unbalanced data set that is the operational data for Air Pressure System (APS) Failure at Scania trucks. The settings and results of this experiment are as follows.

Table 7: The network model for APS Failure dataset

| Input 171-dimensional examples |
| --- |
| (171, 100) linear layer, ReLU |
| (100, 100) linear layer, Tanh |
| (100, 2) linear layer |

Table 8: Description of APS Failure dataset

| Classes | Train Set | Validation Set | Test Set |
| --- | --- | --- | --- |
| Negative | 10000 | 1000 | 5 |
| Positive | 10 | 1000 | 5 |

Table 9: Results with different metrics on APS Failure dataset

| Methods | Accuracy | Marco-F1 score | G-means |
| --- | --- | --- | --- |
| Baseline | $50.00 \pm 0.00$ | $33.33 \pm 0.00$ | $0.00 \pm 0.00$ |
| Proportion | $93.29 \pm 2.59$ | $93.27 \pm 2.63$ | $93.17 \pm 2.78$ |
| Two-phase | $93.19 \pm 0.87$ | $93.19 \pm 0.87$ | $93.15 \pm 0.89$ |
| Hu et al.'s | $94.09 \pm 0.94$ | $93.35 \pm 1.73$ | $93.28 \pm 1.79$ |
| Hu et al.'s+R | $93.98 \pm 1.78$ | $94.65 \pm 0.76$ | $94.65 \pm 0.76$ |
| **Ours.** | $\mathbf{94.49 \pm 0.70}$ | $\mathbf{95.10 \pm 0.66}$ | $\mathbf{94.86 \pm 0.58}$ |

**Dataset and Models.** APS Failure dataset is from UCI Machine Learning Repository (Dua & Graff, 2017). This dataset contains 76000 examples. We randomly construct the train, validation and test data sets from the original data set. These sub-data sets are described in Table 8. The train set consists of 10000 negative examples and 10 positive examples, which is extremely imbalanced, and its imbalance rate reaches 1:1000. The validation and test sets are balanced data sets, and their example sizes for each class are 5 and 1000 respectively. In addition, APS Failure dataset dataset has 171 features, and we replace the missing feature values with the average of the examples in the same class. Finally, we scale the all feature values to $[0, 1]$. The model for APS Failure dataset is shown in Table 7 and is a simple 3-layer FCN.

**Other Settings.** During training, we use Adam optimization with an initial learning rate of 1e-2 and set the batch size to 128. Other hyperparameter settings are the same as the text classification in Section 4.1.

**Results.** The results of different metrics on APS Failure dataset are shown in Table 9. There are three main observations. First, our method achieves the best score in all metrics. It indicates that our method has comprehensive advantages compared with other methods. Second, the score rankings of the six methods are almost consistent among these metrics. It shows that the evaluation of these metrics on a balanced test set is similar. Third, our method performs well on a large-scale data set. It demonstrates that our method is also effective on large-scale data.

## A.2 CONVERGENCE PROOF OF OUR METHOD

This section firstly provides a proof of the convergence of the learning method with a constraint in Section 3.1, and then we prove the convergence of the combination method in Section 3.2.

**Definition 1.** *A function $f(x) : R^d \rightarrow R$ is said to be Lipschitz-smooth with constant L if*

$$\| \nabla f(x) - \nabla f(y) \| \leq L \| x - y \|, \forall x, y \in R^d$$

**Definition 2.** *A function $f(x)$ has $\sigma$-bounded gradients if*

$$\| \nabla f(x) \| \leq \sigma, \ \forall x \in R^d$$

**Theorem 1.** *Suppose the validation loss function $\mathcal{L}_{val}$ is Lipschitz-smooth with constant L, and the training loss function $f_i$ corresponding to the example $x_i$ has $\sigma$-bounded gradients and the Hessian matrix H, namely, the second derivative of $\mathcal{L}_{train}$ with respect to $\theta$, is bounded by $\rho$. Let the learning rate $\eta'_{w_t}$ satisfies $\eta'_{w_t} \leq \frac{2}{LN^2\sigma^2\rho^2}$. Then, after each iteration of the model parameter $\theta$, the validation loss always decreases. More specifically,*

$$\mathcal{L}_{val}(\theta_{t+1}) \leq \mathcal{L}_{val}(\theta_t) \tag{10}$$

**Proof.** The validation loss function $\mathcal{L}_{val}$ is Lipschitz-smooth, so we have

$$\mathcal{L}_{val}(\theta_{t+1}) \leq \mathcal{L}_{val}(\theta_t) + (\nabla_\theta \mathcal{L}_{val})^T \Delta\theta + \frac{L}{2} \| \Delta\theta \| \tag{11}$$

Let $v = (\nabla_\theta \mathcal{L}_{val})^T \Delta\theta + \frac{L}{2} \| \Delta\theta \|$. We can see that only $v \leq 0$, there is $\mathcal{L}_{val}(\theta_{t+1}) \leq \mathcal{L}_{val}(\theta_t)$. Then, substituting the $\theta$ update formula: $\Delta\theta = \nabla_w \theta \, (w_{t+1} - w_t)$ and the $w$ update formula: $w_{t+1} - w_t = \eta'_{w_t} (\nabla_w \theta)^T \nabla_\theta \mathcal{L}_{val}$ from Section 3.1 into $v$, we have

$$v = (\nabla_\theta \mathcal{L}_{val})^T \nabla_w \theta \, (w_{t+1} - w_t) + \frac{L}{2} \| \eta'_{w_t} \nabla_w \theta \, (w_{t+1} - w_t) \| \tag{12}$$

$$= -\eta'_{w_t} (\nabla_\theta \mathcal{L}_{val})^T \nabla_w \theta (\nabla_w \theta)^T \nabla_\theta \mathcal{L}_{val} + \frac{L \eta'_{w_t}{}^2}{2} \| \nabla_w \theta (\nabla_w \theta)^T \nabla_\theta \mathcal{L}_{val} \| \tag{13}$$

$$= (\nabla_\theta \mathcal{L}_{val})^T \nabla_w \theta (\nabla_w \theta)^T \Big[ \frac{L \eta'_{w_t}{}^2}{2} \nabla_w \theta (\nabla_w \theta)^T - \eta'_{w_t} I \Big] \nabla_\theta \mathcal{L}_{val} \tag{14}$$

where $I$ is the identity matrix.

Let $S = \nabla_w \theta (\nabla_w \theta)^T \Big[ \frac{L \eta'_{w_t}{}^2}{2} \nabla_w \theta (\nabla_w \theta)^T - \eta'_{w_t} I \Big]$, so $v = (\nabla_\theta \mathcal{L}_{val})^T S \, \nabla_\theta \mathcal{L}_{val}$. Next, we prove that $S$ is a semi-negative definite matrix such that $v \leq 0$.

We observe that in $S$, the term $\nabla_w \theta (\nabla_w \theta)^T$ is a symmetric and positive semi-definite matrix. We use $A$ to denote this term and define its eigendecomposition as $A = P diag(\lambda) P^{-1}$ where $\lambda$ is a vector composed of eigenvalues and $\lambda_i \geq 0$ for all $i$. Substituting this eigendecomposition into $S$, we have

$$S = P diag(\lambda) P^{-1} (\frac{L \eta'_{w_t}{}^2}{2} P diag(\lambda) P^{-1} - \eta'_{w_t} I) \tag{15}$$

$$= P diag(\frac{L \eta'_{w_t}{}^2}{2} \lambda * \lambda - \eta'_{w_t} \lambda) P^{-1} \tag{16}$$

where $*$ represents the hadamard product.

Therefore, in order to make $S$ a semi-negative definite matrix, $\eta'_{w_t}$ must satisfies $\frac{L \eta'_{w_t}{}^2}{2} \lambda_i^2 - \eta'_{w_t} \lambda_i \leq 0$ for all $i$, namely, $0 \leq \eta'_{w_t} \leq \frac{2}{L \lambda_i}$. Let $\lambda_{max} = max_i(\lambda_i)$, so finally $\eta'_{w_t}$ must satisfies

$$0 \leq \eta'_{w_t} \leq \frac{2}{L \lambda_{max}} \tag{17}$$

Further, we estimate the boundary of the scalar value $\lambda_{max}$. In Section 2, we define a matrix $F = (\nabla f_1, ..., \nabla f_N)$, where $f_i$ is the training loss function and $N$ is the number of the training examples. Since $f_i$ has $\sigma$-bounded gradients, we can obtain $\| F \| \leq N\sigma$. Substituting $\nabla_w \theta$ in Eq. 7 into $A$, we have

$$\| A \| = \| \nabla_w \theta (\nabla_w \theta)^T \| = \| diag(h) F F^T diag(h) \| \tag{18}$$

$$\leq \| diag(h) \| \| F \| \| F^T \| \| diag(h) \| = N^2 \sigma^2 \rho^2 \tag{19}$$

Since $\lambda_{max}$ is the eigenvalue of $A$, we can obtain

$$\lambda_{max} \leq \| A \| = N^2 \sigma^2 \rho^2 \tag{20}$$

Therefore, combining Eq. 17 and Eq. 20, the satisfying range of $\eta'_{w_t}$ is

$$0 \leq \eta'_{w_t} \leq \frac{2}{L N^2 \sigma^2 \rho^2} \tag{21}$$

This finishes our proof for Theorem 1. $\square$

**Theorem 2.** *Suppose the validation loss function $\mathcal{L}_{val}$, the training loss function $f_i$ and the learning rate $\eta'_{w_t}$ satisfies Theorem 1 conditions. Same as Algorithm 1, $t$ is denoted as the time step where the algorithm successively uses Hu et al. and our methods to update $\theta$, and let $t'$ represent the time step inside time-step $t$ and the algorithm only apply our method to update $\theta$. The range of $t'$ is $[0, ..., T_2 - 1]$. Then the validation loss always decreases after the $t$-th iteration, namely,*

$$\mathcal{L}_{val}(\theta_{t+1}) \leq \mathcal{L}_{val}(\theta_t) \tag{22}$$

**Proof.** The $\theta_{t'=0}$ is the updated parameter through the method (Hu et al.). According to the convergence theorem in the paper (Ren et al., 2018; Hu et al., 2019), we can obtain

$$\mathcal{L}_{val}(\theta_t) \geq \mathcal{L}_{val}(\theta_{t'=0}) \tag{23}$$

After that, we use our method to update the parameter $T$ times. According to the Theorem 1, we have

$$\mathcal{L}_{val}(\theta_{t'=0}) \geq \mathcal{L}_{val}(\theta_{t'=1}) ... \geq \mathcal{L}_{val}(\theta_{t'=T_2-1}) \tag{24}$$

Here, the algorithm completes the update of the parameter in time-step $t$, namely, $\theta_{t'=T_2-1} = \theta_{t+1}$. Combining Eq. 23 and Eq. 24, we can obtain

$$\mathcal{L}_{val}(\theta_{t+1}) \leq \mathcal{L}_{val}(\theta_t) \tag{25}$$

This finishes our proof of Theorem 2. □

### A.3 CONVERGENCE RATE OF OUR METHOD

This section provides a proof of the convergence rate of the learning method with a constraint in Section 3.1.

**Theorem 3.** *Suppose the validation loss function $\mathcal{L}_{val}$, the training loss function $f_i$ and the learning rate $\eta'_{w_t}$ satisfies Theorem 1 conditions. Then the learning method in Section 3.1 achieves $\| \nabla_{\theta_t}\mathcal{L}_{val} \| \leq \epsilon$ in $\mathcal{O}(\frac{1}{\epsilon})$ steps, namely,*

$$\min_{0 < t < T} \| \nabla_{\theta_t}\mathcal{L}_{val} \| \leq \frac{C}{T} \tag{26}$$

*where $C$ is some constant*

**Proof.** According to Eq. 11 and Eq. 14 in Theorem 1, we can obtain

$$\mathcal{L}_{val}(\theta_{t+1}) - \mathcal{L}_{val}(\theta_t) \leq (\nabla_{\theta_t}\mathcal{L}_{val})^T S_t \nabla_{\theta_t}\mathcal{L}_{val} \tag{27}$$

where $S_t$ is the matrix $S$ at time-step $t$ and $S$ is defined in Theorem 1.

Then we have

$$\sum_{t=0}^{T} (\nabla_{\theta_t}\mathcal{L}_{val})^T S_t \nabla_{\theta_t}\mathcal{L}_{val} \geq \mathcal{L}_{val}(\theta_{T+1}) - \mathcal{L}_{val}(\theta_0)$$
$$\geq \mathcal{L}_{val}(\theta^*) - \mathcal{L}_{val}(\theta_0) \tag{28}$$

where $\mathcal{L}_{val}(\theta^*)$ is the minimum of function $\mathcal{L}_{val}(\theta)$. Then, we can observe that there exist a time-step $0 \leq \tau \leq T$ such that,

$$T(\nabla_{\theta_\tau}\mathcal{L}_{val})^T S_\tau \nabla_{\theta_\tau}\mathcal{L}_{val} \geq \mathcal{L}_{val}(\theta^*) - \mathcal{L}_{val}(\theta_0) \tag{29}$$

We have proved that $S_\tau$ is a semi-negative definite matrix. According to Eq. 16, we have

$$S_\tau = Pdiag(\lambda')P^{-1} = Pdiag(\lambda')P^T \tag{30}$$

where $\lambda'$ is a vector composed of eigenvalues and $\lambda'_i \leq 0$ for all $i$. $P$ is the matrix composed of eigenvectors of $A$ in Theorem 1 and $P^{-1} = P^T$ because $A$ is a symmetric matrix. Substituting Eq. 30 into Eq. 29, we have

$$\mathcal{L}_{val}(\theta^*) - \mathcal{L}_{val}(\theta_0) \leq T(\nabla_{\theta_\tau}\mathcal{L}_{val})^T P diag(\lambda')P^T \nabla_{\theta_\tau}\mathcal{L}_{val} \tag{31}$$

$$= T(P^T \nabla_{\theta_\tau}\mathcal{L}_{val})^T diag(\lambda')P^T \nabla_{\theta_\tau}\mathcal{L}_{val} \tag{32}$$

$$\leq T(P^T \nabla_{\theta_\tau}\mathcal{L}_{val})^T \lambda'_{max} I P^T \nabla_{\theta_\tau}\mathcal{L}_{val} \tag{33}$$

$$= T\lambda'_{max} \parallel \nabla_{\theta_\tau}\mathcal{L}_{val} \parallel \tag{34}$$

We can regard Eq. 32 as the quadratic form of the diagonal matrix $diag(\lambda')$, and we scale all the eigenvalues $\lambda'_i$ to $\lambda'_{max}$ where $\lambda'_{max} = \max_i \lambda'_i$, so that we obtain the inequality in Eq. 33.

So we have

$$\parallel \nabla_{\theta_\tau}\mathcal{L}_{val} \parallel \leq \frac{1}{T}\frac{\mathcal{L}_{val}(\theta^*) - \mathcal{L}_{val}(\theta_0)}{\lambda'_{max}} = \frac{C}{T} \tag{35}$$

where $C = \frac{\mathcal{L}_{val}(\theta^*) - \mathcal{L}_{val}(\theta_0)}{\lambda'_{max}}$ is a constant independent of $T$.

Therefore, we can obtain $\min_{0<t<T} \parallel \nabla_{\theta_t}\mathcal{L}_{val} \parallel \leq \parallel \nabla_{\theta_\tau}\mathcal{L}_{val} \parallel \leq \frac{C}{T}$. It means that our method can achieve $\min_{0<t<T} \parallel \nabla_{\theta_t}\mathcal{L}_{val} \parallel \leq \mathcal{O}(\frac{1}{T})$ in $T$ steps. This finishes our proof of Theorem 3. $\qquad\square$

