# OpenReview forum: "Improving the Accuracy of Learning Example Weights for Imbalance Classification"
_ICLR.cc/2022/Conference — ICLR 2022 Poster_

### Official Review · Reviewer_auNr · 2021-10-30

**Correctness:** 4
**Technical Novelty And Significance:** 3
**Empirical Novelty And Significance:** 3
**Recommendation:** 6
**Confidence:** 4

**Main Review:**

The authors start by explaining the process of learning from a class imbalance dataset. In such a scenario, different weights are given to the different training examples. The model objective is to identify the optimal model that minimizes the example-weighted loss of the training set. The meta objective is to give the best weight to the training example so that the model minimizes the loss of the balanced validation set.

The authors first propose a method to learn theta and w with a constraint, which can accurately optimize both parameters. Next they combine this with a method previously proposed to boost the performance further. The authors start by noting that the optimal model parameter theta* and w satisfy the constraint F(theta*)w = 0. This constraint gives a precise relationship between these two parameters. From this equation they identify the gradient to be related to the inverse of the Hessian of F(theta). This helps them derive the iterative update rules for both theta and w which satisfy the constraint.

However, this solution only satisfies the meta objective as described before. This is not optimal for the model objective. As such, they combine their approach with the one proposed by Hu et al., to get the best results.

The authors then show the efficacy of the proposed method under different experimental setups. This shows that the proposed combined approach can be very effective under different imbalance ratios.



**Summary Of The Paper:**

The authors propose an approach to tackle the class imbalance problem present widely in the machine learning domain. For this purpose, they first propose a mechanism to precisely learn the relationship between the weights and the trained model in the model objective. This allows the weights and models to be optimized more accurately. They then combine this process with the mechanism proposed by Hu et al., which helps the model learn the model objective better. Finally, they show the efficacy of their proposed method through experiments.

**Summary Of The Review:**

The paper is relatively smooth to read. However, the descriptions can sometimes be very verbose (e.g., introduction) and thus there is some room for improvement there. Overall the approach seems justified and the results are promising. However, the improvements can be minimal in some cases from the baselines as shown by the experimental results. Another potential issue can be the low volume of training data. I would have liked to see more larger datasets being used here as in practice much bigger datasets are the use cases where the class imbalance problem pops up.

---

> ### Author Response · Authors · 2021-11-22
> **Response to Reviewer auNr**
>
> [Q1]: However, the improvements can be minimal in some cases from the baselines as shown by the experimental results.
>
> [A1]: In some cases, our method indeed has only slight improvements compared to the method of Hu et al. It shows that although the method (Hu et al.) uses the local approximate method to optimize the parameters, it is still possible to achieve good performance on some data sets. However, the method (Hu et al.) does not work on severely unbalanced data or in difficult classification situations, such as SST-5 dataset. In contrast, our method has significant improvements in these difficult classification tasks.
>
> [Q2]: Another potential issue can be the low volume of training data. I would have liked to see more larger datasets being used here as in practice much bigger datasets are the use cases where the class imbalance problem pops up.
>
> [A2]: Thanks for your suggestion. We have added a large-scale dataset and evaluated our method on this dataset in Appendix A.1. The training set contains 10000 negative examples and 10 positive examples. The results in Table 9 show that our method has consistent improvement in different metrics. It demonstrates that our method is also effective on a large-scale dataset.

---

> > ### Comment · Reviewer_auNr · 2021-11-23
> > **Response to Author Feedback**
> >
> > I thank the authors for their feedback. After reading the other reviews and author feedback I have decided not to update the score at the moment.

---

### Official Review · Reviewer_fpgG · 2021-11-02

**Correctness:** 4
**Technical Novelty And Significance:** 3
**Empirical Novelty And Significance:** 3
**Recommendation:** 8
**Confidence:** 3

**Main Review:**

Strong points:
The problem is very important. The solution is novel from theoretical perspective, and it is evident in the experiments as well.
The competing methods does not only contain previous published work but also include its variants to improve the performance, which make the proposed method further attractive.

The paper is well written. Both the proposed method and experiments are clearly explained.

Weak points:
According to Algorithm 1, the proposed method needs O(T^2) to converge, where T is the number of iterations. However, if I understand correctly, the previous method requires O(T) for convergence. Will the proposed method need much more time to converge in practice?

In Eq.(7), the authors uses diagonal approximation to replace the inverse of Hessian. Would other methods, e.g., Quasi-Newton’s method, give a more accurate approximation without too much cost? Please comment on this.

It might be good to include the cost of such approximation in the future work when using a small model. The proposed method may also beneficial other shallow models as well.


**Summary Of The Paper:**

This paper proposes a new approach to learn sample weights aiming to solve imbalance classification problem. The problem is challenging because the learnable model parameter and sample weights are coupled and cannot be directly optimized together. The previous method learns the parameter and weights in an alternative fashion, which solves the problem approximately. The authors argues that there exists a constraint and allow both parameters to be learned together thus this paper proposed a new algorithm combined with previous method to learn the sample weights. Experimental results shows that the proposed method outperforms other competing method especially for the extreme imbalance cases (1:100).

**Summary Of The Review:**

As many real-world datasets naturally forms in an imbalanced form, the solution provided in this paper motivated from theoretical side and is strongly proved empirically by comparing with other methods on different datasets.

---

> ### Author Response · Authors · 2021-11-22
> **Response to Reviewer fpgG**
>
> [Q1]: According to Algorithm 1, the proposed method needs O(T^2) to converge, where T is the number of iterations. However, if I understand correctly, the previous method requires O(T) for convergence. Will the proposed method need much more time to converge in practice?
>
> [A1]: Thanks for pointing out this issue. We have given proofs of the convergence rate of the learning method in section 3.1 and also explain the convergence rate of Algorithm 1 in the last paragraph of section 3.2. More specifically, if the previous method requires T steps for convergence, then Algorithm 1 needs $O(T^{\frac{3}{2}})$ steps to converge. In practice, our proposed method does require more time to train the model but the time consumption is affordable.
>
> [Q2]: In Eq.(7), the authors uses diagonal approximation to replace the inverse of Hessian. Would other methods, e.g., Quasi-Newton’s method, give a more accurate approximation without too much cost? Please comment on this.
>
> [A2]: In the paper, our method is applied to the deep learning models and the number of model parameters is large. Therefore, we need a fast, low-cost and effective method. The quasi-Newton method still requires a relatively large time and memory cost compared with first-order optimization methods. For example, L-BFGS, a commonly used quasi-Newton method, needs to store the first-order gradient information of the previous m iterations (commonly m>=5), and its storage consumption is still unbearable for deep neural networks with millions of parameters. Moreover, stochastic optimization is often used in deep learning, but for quasi-Newton methods, it may lead to an erroneous approximation of Hessian [1] and make the learning process unstable.
> Compared with other methods, the diagonal approximation is linear complexity in both time and memory and is as efficient as the first-order optimization methods. In addition, the experimental results indicate that the diagonal approximation is effective in varieties of settings.
>
> [1] Richard H Byrd, Gillian M Chin, Will Neveitt, and Jorge Nocedal. On the use of stochastic hessian information in optimization methods for machine learning. SIAM Journal on Optimization, 21(3):977–995, 2011.
>
> [Q3]: It might be good to include the cost of such approximation in the future work when using a small model. The proposed method may also beneficial other shallow models as well.
>
> [A3]: Thanks so much for the suggestion. Other more accurate approximation methods, such as the quasi-Newton method, may have better results when using small models. In future work, we will consider the efficiency and effectiveness of the inverse Hessian approximation, especially in shallow models.

---

### Official Review · Reviewer_yKjB · 2021-11-02

**Correctness:** 3
**Technical Novelty And Significance:** 3
**Empirical Novelty And Significance:** 3
**Recommendation:** 6
**Confidence:** 4

**Main Review:**

The proposed method provides some useful results, but there are some questions needed to be answered.
1.  For the learned θ in Eq. (1), why it is an unbiased θ, and you need to give more details.
2.  Eq. (6) is used as a functional relationship between θ and w, but you tell us little about how to perform this constraint in the final objective function and optimized process because this is important for your model.
3.  In Section 3.2, you combine your model with another algorithm in section 2, which can guarantee that θ is optimal for Eq.1 and Eq.2. Intuitively, this model has significant improvement in varieties of settings, but whether you give a simple mathematical proof.
4. In your experimental settings, note that you use a small balanced validation set, why? Since your setting is designed for the imbalanced classification, and whether you need to utilize an unbalanced validation set.
5.  Since the four datasets used in this paper essentially include two kinds of datasets, thus you need to provide more results for different types of datasets.
6.  There are various metrics (e.g., Acc, Recall, F_score, G-mean) for evaluating the imbalanced classification, however, you only use Acc in your paper. This is not comprehensive to show the advantages of the proposed method in the imbalanced classification.
7.	Several mistakes need to be corrected. For example, the model parameters θ obtained by this method is biased. “parameters” should be “parameter”. Therefore, I suggest that you should check your paper carefully to avoid similar mistakes.


**Summary Of The Paper:**

They propose a novel mechanism of learning with a constraint, which can accurately train the weights and model. Then, they propose a combined method of our learning mechanism and the work by Hu et al., which can promote each other to perform better.

**Summary Of The Review:**

In a word, the idea of the proposed model is useful to obtain a better performance, however, several detail issues are not clear in the proposed model, and the experimental results are not sufficient to verify the effectiveness of the proposed method.

---

> ### Author Response · Authors · 2021-11-22
> **Response to Reviewer yKjB**
>
> [Q1]: For the learned $θ$ in Eq. (1), why it is an unbiased $θ$, and you need to give more details.
>
> [A1]: Thanks for pointing out this for us. Sorry for the ambiguous use of the “unbiased” term in a non-standard way. The $θ$ learned in Eq. (1) is not the statistically unbiased estimate, but a parameter that is fair to the minority and majority classes and is not biased towards the majority class.  We have corrected the use of “unbiased” term in Section 2.
>
> [Q2]: Eq. (6) is used as a functional relationship between $θ$ and $w$, but you tell us little about how to perform this constraint in the final objective function and optimized process because this is important for your model.
>
> [A2]: Our main idea is to use the constraint in Eq. (6) to obtain a precise differential relationship between $θ$ and $w$, so we can use this differential to calculate the gradient $\nabla_{w} L_{val}$  to update $w$ ($L_{val}$ is the final objective function).  In addition, we also use this differential to calculate the new $θ$ corresponding to the updated $w$. Therefore, in the optimization process, $θ$ and $w$ all satisfy the constraint. We have added above two details in Section 3.1. First, we clearly point out that the derivative of $θ$ with respect to $w$ is obtained from the constraint in Eq. (6). Second, we make it clear that in the optimization process, the calculation of new $θ$ is to use the differential in Eq. (6) and meet the constraint of Eq. (6).
>
> [Q3]: In Section 3.2, you combine your model with another algorithm in section 2, which can guarantee that $θ$ is optimal for Eq.1 and Eq.2. Intuitively, this model has significant improvement in varieties of settings, but whether you give a simple mathematical proof.
>
> [A3]: Thanks for your suggestion. We have provided the proofs of the convergence in Theorem 1 and 2 in Appendix A.2. Theorem 2 guarantees that the parameter $θ$ can minimize the loss on the validation set (local minimum), namely, is optimal for Eq.2. In addition, when $θ$ converges, the $θ$ update proposed by Hu et al. [2] in the combined method also guarantees $θ$ is optimal for Eq.1. Therefore, our method can guarantee that $θ$ is optimal for Eq.1 and Eq.2.
>
> [Q4]: In your experimental settings, note that you use a small balanced validation set, why? Since your setting is designed for the imbalanced classification, and whether you need to utilize an unbalanced validation set.
>
> [A4]: In Section 2, we show that the optimal $w$ is to make the model parameter $θ$ learned from Eq.1 minimize the loss on a balanced validation set. It means that the best $w$ is to make the model perform well on a balanced data set. Therefore, our algorithm requires a balanced validation set for training w and a large but unbalanced train set for training $θ$. Many similar works have been based on this idea [1], [2]. In order to enable readers to understand our algorithm better, we have added details and pointed out that $θ$ in Eq.1 is learned on unbalanced training set and $w$ in Eq.2 is learned on balanced validation set.
>
> [Q5]: Since the four datasets used in this paper essentially include two kinds of datasets, thus you need to provide more results for different types of datasets.
>
> [A5]: Thanks for your suggestion. We have added a new data set and evaluated our method with different metrics in Appendix A.1. The data set is from UCI Machine Learning Repository and is the operational data for Air Pressure System (APS) Failure at Scania trucks. The experimental results are shown in Table 9 and demonstrate that our method can also perform well for other types of data.
>
> [Q6]: There are various metrics (e.g., Acc, Recall, F_score, G-mean) for evaluating the imbalanced classification, however, you only use Acc in your paper. This is not comprehensive to show the advantages of the proposed method in the imbalanced classification.
>
> [A6]: Thank you for the suggestion. We have added Macro-F1 score and G-means metrics to the new data set. The experimental results in Table 9 show that our method has consistent improvement in accuracy, Macro-F1 score and G-means. In addition, the score rankings of the six comparison methods are almost consistent among these metrics. It demonstrates that the performance of these metrics on a balanced test set is similar. In practice, many imbalance works only use accuracy on a balanced test set [1], [2]. Due to the limited rebuttal periods, the other datasets in our work are still the same as existing work and we only report their results on accuracy.
>
> [Q7]: “parameters” should be “parameter”
>
> [A7]: Thank you for the suggestion. We have corrected these mistakes in the revised version.
>
> [1] Mengye Ren, Wenyuan Zeng, Bin Yang, and Raquel Urtasun. Learning to reweight examples for robust deep learning.ICML, 2018.
>
> [2] Zhiting Hu, Bowen Tan, Ruslan Salakhutdinov, Tom Mitchell, and Eric P. Xing. 2019. Learning data manipulation for augmentation and weighting. NIPS.

---

### Official Review · Reviewer_XCGn · 2021-11-03

**Correctness:** 3
**Technical Novelty And Significance:** 3
**Empirical Novelty And Significance:** 2
**Recommendation:** 6
**Confidence:** 3

**Main Review:**

Strengths:
-Good review of existing work on the topic
-Formulation of the proposed approach is well motivated
-Extensive experimental analysis
Weaknesses:
-Some concepts in section 2 are not well defined and are used in non-standard ways: for example, what does it mean for the parameters of a model to be biased? What does it mean for a validation dataset to be unbiased?
-The novelty and significance are limited, since the proposed method is similar to the one proposed by Hu et al and the experimental results in most cases show no statistically significant difference in the results (although for one case the difference is very large).
-In the title and abstract they refer to "Imbalance classification" but it should be "Imbalanced classification"



**Summary Of The Paper:**

The paper presents a new method for learning example weights together with the parameters of a deep neural network. The difference between the proposed method and previous work is that they use a constraint to tie together the values of the parameters and the weights as they do the joint optimization through gradient descent. They do extensive experiments using both text and image datasets, with different imbalance ratios and show that the proposed method outperforms the state-of-the-art in terms of accuracy on a (balanced) test set.

**Summary Of The Review:**

The proposed method is well-motivated: it is a small modification to recent related work that seems to be justified mathematically by the constraint argument they give for tying together the optimization of the weights and parameters of the network. The paper is generally well-written but I missed a better setting of the problem in section 2 -- which I find confusing -- especially in the use of the "biased"/"unbiased" terms when referring to parameters and datasets. The experimental section describes the experiments with enough details that they can be reproduced, the experiments are well structured and the accuracy results show a small but consistent improvement in most experiments over the state-of-the-art. Therefore, I believe the contribution of this paper is just enough to warrant acceptance.

---

> ### Author Response · Authors · 2021-11-22
> **Response to Reviewer XCGn**
>
> [Q1]: Some concepts in section 2 are not well defined and are used in non-standard ways: for example, what does it mean for the parameters of a model to be biased? What does it mean for a validation dataset to be unbiased?
>
> [A1]: Thanks for pointing out this issue. We are sorry for the ambiguous use of the “biased”/“unbiased” terms in a non-standard way. We have corrected these corresponding statements in our revision. Specifically, the “biased” model parameter in the paper is not the statistically biased estimate but a parameter that is biased towards the majority class in imbalanced classification. The “unbiased” validation dataset refers to a balanced dataset.
>
> [Q2]: The novelty and significance are limited, since the proposed method is similar to the one proposed by Hu et al and the experimental results in most cases show no statistically significant difference in the results (although for one case the difference is very large).
>
> [A2]: The previous method adopts the local approximation to optimize $w$, namely, only considering the influence of $w$ on the parameter $\theta$ in a short term (in a gradient descent step). It may result in inaccurate learning on $w$. In contrast, our proposed method uses the constraint in Eq. (6) to obtain a precise differential relationship between $θ$ and $w$, and then our method can utilize the relationship to better optimize $w$. Therefore, our method can make the $w$ and $θ$ learning more accurate. This is a significant difference between our method and the previous work. In addition, the experimental results in section 4.3 show that when the imbalance of the datasets is serious, our method has a significant improvement compared with other methods. Especially for the extreme imbalance cases (1:100), the accuracy of our method outperforms the second-best method by more than 5 accuracy points on SST-2 in Table 5 and exceeds the second-best method by more than 6 accuracy points on CIFAR100 in Table 6.
>
> [Q3]: In the title and abstract they refer to "Imbalance classification" but it should be "Imbalanced classification"
>
> [A3]: Thank you for pointing out this issue. We have corrected these mistakes in the revised version.

---

### Public Comment · ~Jun_Shu1 · 2022-02-02
**Related work suggestion**

Hi authors,

Thanks for this interesting work for proposing an improved algorithm to learn the example weights for imbalance classification. I'd like to point a (possibly) missing related work, which may be worthwhile to be discussed in the paper.

The Meta-Weight-Net technique (MW-Net, published in NeurIPS 2019, https://arxiv.org/abs/1902.07379) also provides an improved algorithm to learn the example weights for imbalance classification. We model the weighting scheme as an MLP network with one hidden layer, constituting a universal approximator to almost any continuous functions, making the method able to fit a wide range of weighting functions including the example weights for imbalance classification. The major difference with L2RW (Ren et al., ICML, 2018) is that the weights are implicitly learned there, without an explicit weighting function. This, however, might lead to unstable weighting behavior during training and unavailability for generalization. In contrast, with the explicit yet simple Meta-Weight-Net, our method can learn the weight in a more stable way.

I believe that citing MW-Net won't devalue yours at all. If you want, I will be happy to discuss further.

Thank you!

---

### Decision · Program_Chairs · 2022-01-20

**Decision:**

Accept (Poster)

**Comment:**

To solve imbalance classification problem, this paper proposes a method to learn example weights together with the parameters of a neural network. The authors proposed a novel mechanism of learning with a constraint, which allows accurate training of the weights and model at the same time. Then they combined this new learning mechanism and the method by Hu et al. (2019), and demonstrated its usefulness in extensive experiments.

I would like to thank the authors for their detailed feedback to the initial reviews, which clarified most of the unclear points in the manuscript. Overall the paper is well written and the effectiveness was demonstrated in experiments. Since the contribution is valuable to ICLR2022, I suggest its acceptance.